# Investigation of Vacancy-Ordered Double Perovskite Halides A_2_Sn_1−x_Ti_x_Y_6_ (A = K, Rb, Cs; Y = Cl, Br, I): Promising Materials for Photovoltaic Applications

**DOI:** 10.3390/nano13202744

**Published:** 2023-10-11

**Authors:** Wen Chen, Gang Liu, Chao Dong, Xiaoning Guan, Shuli Gao, Jinbo Hao, Changcheng Chen, Pengfei Lu

**Affiliations:** 1School of Science, Xi’an University of Architecture and Technology, Xi’an 710055, China; chenwen@xauat.edu.cn (W.C.); gaoshuli@xauat.edu.cn (S.G.); jbhao@xauat.edu.cn (J.H.); 2School of Electronic Engineering, Beijing University of Posts and Telecommunications, Beijing 100876, China; 3School of Integrated Circuits, Beijing University of Posts and Telecommunications, Beijing 100876, China; chaodong@bupt.edu.cn (C.D.); guanxn@bupt.edu.cn (X.G.); lupengfei@bupt.edu.cn (P.L.)

**Keywords:** vacancy-ordered, double perovskite, density functional theory, toxic-free, solar photovoltaics

## Abstract

In the present study, the structural, mechanical, electronic and optical properties of all-inorganic vacancy-ordered double perovskites A_2_Sn_1−x_TixY_6_ (A = K, Rb, Cs; Y = Cl, Br, I) are explored by density functional theory. The structural and thermodynamic stabilities are confirmed by the tolerance factor and negative formation energy. Moreover, by doping Ti ions into vacancy-ordered double perovskite A_2_SnY_6_, the effect of Ti doping on the electronic and optical properties was investigated in detail. Then, according to the requirement of practical applications in photovoltaics, the optimal concentration of Ti ions and the most suitable halide element are determined to screen the right compositions. In addition, the mechanical, electronic and optical properties of the selected compositions are discussed, exhibiting the maximum optical absorption both in the visible and ultraviolet energy ranges; thus, the selected compositions can be considered as promising materials for application in solar photovoltaics. The results suggest a great potential of A_2_Sn_1−x_Ti_x_Y_6_ (A = K, Rb, Cs; Y = Cl, Br, I) for further theoretical research as well as experimental research on the photovoltaic performance of stable and toxic-free perovskite solar cells.

## 1. Introduction

The global energy demand keeps increasing owning to the development of the international economy, and this motivates researchers to seek alternatives to conventional fossil fuels, which are currently the main energy source but with limited reserves. Among the alternatives, solar energy is both clean and endless, but most importantly, it is conveniently available whenever and wherever possible. Therefore, it is considered as the long-term and economic substitute for fossil fuel. As the leading source of renewable energy, the performance of materials that are employed to use this energy source depends mostly on their band gap and feasibility of application. In this regard, considerable efforts have been made to find environmentally friendly and cost-effective technological systems and utilize the energy spectrum for solar cells and thermoelectric generators [1,2,3,4]. At present, the critical challenge is to find suitable materials, and much research has already been conducted to exploit their capability in energy conversion [5,6,7].

Hybrid organic–inorganic materials, methylammonium lead halides [8,9], in perovskite structures have proved successful as light absorbers in perovskite solar cells (PSCs) due to their unique optical and electronic properties, such as strong optical absorption, tunable band gap and large carrier diffusion length. The efficiency of single-junction PSCs were increased noticeably from 3.8% in 2009 [5] to 25.5% in 2022 [10,11], which exceeded that of many conventional semiconductor absorbers. Moreover, Catchpole et al. reported an efficiency of 26.4% of solar cells by the tandem of PSCs and silicon-based solar cells [12], while the predicted theoretical photoelectric conversion efficiency(PCE) of PSCs is 31.4% [13]. Although hybrid organic–inorganic perovskite-based optoelectronic devices have been extensively researched, they are still far from commercialization since two critical problems still exist and are urgent to be overcome. One is the intrinsic toxicity of lead possibly leading to environmental pollution. The other is the poor structural stability of the volatile and hygroscopic organic cations to ambient environments, such as oxygen [14,15], humidity [16,17,18], light [19,20], temperature or thermal stress [21,22]. As a result, it is important to find all-inorganic lead-free perovskite materials for photovoltaics and other optoelectronic applications [23,24,25,26], and concerted efforts have been made to design new lead-free and stable halide perovskite for solar cells as well as other optoelectronic applications [27,28,29].

Among the suitable alternatives, the all-inorganic vacancy-ordered double perovskites of type A_2_BX_6_ have attracted much attention for application in photovoltaic technology due to their long-term stability and environmentally friendly properties [30,31]. They are a family of defect-ordered structural derivatives of the archetypal perovskite structure, and their structure is obtained by doubling the conventional perovskite ABX_3_ unit cell along the three different crystallographic axes and then removing every other B-site cation. The remaining B-site cations form an ordered face-centered cubic arrangement. Since the [BX_6_] octahedrons are not connected by alternating B-X bonds, and each anion is coordinated to only one B-site cation, this structure can also be considered as an antifluorite arrangement of anionic isolated [BX_6_] octahedrons bridged by A-site cations, to which these materials are often referred. Compared with the conventional ABX_3_ structure, the vacancy-ordered double perovskite A_2_BX_6_ structure has a tetravalent B-cation, instead of a bivalent B-cation in ABX_3_, which endows it with enhanced air and moisture stability under ambient conditions [23,30]. Although vacancy-ordered double perovskite materials have been studied in the literature, previous research almost exclusively focused on their structural and dynamic behaviors. Their properties relevant to specific potential applications in photovoltaics and other optoelectronics have been seldom studied systematically.

In order to further optimize the properties of materials, substitutional alloying or doping has been demonstrated to be a very effective strategy. Doping is at present an important method to improve the properties of perovskite materials, and vacancy-ordered double perovskite can be readily doped with different ions at all three sites to obtain the desired electronic and optical properties, especially in the sixfold coordinated tetravalent cation site. In addition, there are a large variety of tetravalent ions that may be introduced into the structure; thus, the compositional modification at the B-site of vacancy-ordered double perovskite A_2_BX_6_ may cause the largest flexibility in optical and electronic properties.

As the typical representation of vacancy-ordered double perovskite A_2_SnY_6_, Cs_2_SnI_6_ was initially applied as a hole transport material for solar cells and attained ~8% efficiency [24]. After that, much attention has been paid to its research. Recent research on A_2_SnY_6_ has provided new prospects for a stabilized and environmentally friendly solar cell material [32,33]. Cs_2_SnCl_6_, Cs_2_SnBr6 and Cs_2_SnI_6_ exhibit stable crystal structures at ambient temperature with a direct band gap semiconducting character and were found suitable for dye-sensitized solar cells [23]. However, there is still much room for improvement in the future; therefore, researchers are seeking different variants by compositional engineering to further enhance the power conversion efficiency of A_2_SnY_6_ perovskites.

In this work, we doped Ti ions into the B-site of a vacancy-ordered A_2_SnY_6_ double perovskite structure, and then explored the structural, electronic and optical properties of all-inorganic vacancy-ordered double perovskites A_2_Sn_1−x_Ti_x_Y_6_ (A = K, Rb, Cs; Y = Cl, Br, I) by density functional theory. The effect of Ti doping on the electronic and optical properties is investigated and the optimal concentration of Ti ions and the most suitable halide element are determined to screen the right compositions for practical applications in photovoltaics. The mechanical, electronic and optical properties of the selected compositions are discussed in detail. Our results are expected to provide a theoretical basis for the development of stable and toxic-free perovskite solar cells.

## 2. Computational Methods

All the calculations in this work were performed using the Vienna ab initio simulation package (VASP). The projector-augmented wave (PAW) method was applied to describe the interactions between the core and valence electrons. The electronic exchange and correlation energy was evaluated by using the generalized gradient approximation (GGA) in the form of Perdew–Burke–Ernzerhof (PBE) functional. The cut-off energy of 550 eV was used. The convergence criterion was set to 10^−5^ eV in energy, and all atomic positions and crystal structures involved in this work were fully relaxed with the force convergence at 0.01 eV/Å. During the optimizing geometry and static energy calculations, the Monkhorst–Pack k-point was chosen as 4 × 4 × 4. The elastic constants *C_ij_* were calculated by the strain–stress relationship with finite differences implemented in the VASP code. 

## 3. Results and Discussions

### 3.1. Structural Properties of A_2_Sn_1−x_Ti_x_Y_6_ (A = K, Rb, Cs; Y = Cl, Br, I)

The structural and thermodynamic stabilities of A_2_Sn_1−x_Ti_x_Y_6_ (A = K, Rb, Cs; Y = Cl, Br, I) are necessary for energy devices. Doping concentrations of Ti^4+^ by face-center substitution with x = 0, 0.25, 0.5, 0.75 and 1 were considered in order to investigate the effect of Ti^4+^ doping. The optimized crystal parameters of A_2_Sn_1−x_Ti_x_Y_6_ (A = K, Rb, Cs; Y = Cl, Br, I) are listed in Appendix A and the optimized crystal structures are plotted in Figure 1. According to Appendix A, the lattice parameter and cell volume increase gradually with the increase in the atomic radius of A-site for the same B-site cation and Y-site anion. For the same A-site and B-site cations, the lattice parameter and cell volume increase gradually with the increase in the atomic radius of Y-site. When the A-site cation and Y-site anion are fixed, the lattice parameter and cell volume decrease monotonously with the increase in the concentration of Ti^4+^, which indicates the contraction of the cell. Among all A_2_Sn_1−x_Ti_x_Y_6_ perovskites, K_2_TiCl_6_ has the lowest lattice parameter, at 10.025 Å, and Cs_2_SnI_6_ has the highest lattice parameter, at 12.056 Å, which is very close to the reported values of 9.79 and 12.04 Å, respectively [34,35].

The Goldschmidt tolerance factor t is used to evaluate whether the cation can be replaced in a perovskite material and predict the structural stability of perovskite materials, which is expressed by the following equation:(1)$$t=\frac{r_{A}+r_{Y}}{\sqrt{2}(r_{Sn/Ti}+r_{Y})}$$
where *r*_A_ and *r*_Y_ are the Shannon’s radii of A and Y atoms, respectively, and
(2)$$r_{Sn/Ti}=(1-x)r_{Sn}+xr_{Ti}$$

When the Goldschmidt’s tolerance factor is in the range of 0.8–1.1, the structure of the materials is generally formed [36,37]. Our calculated results of the tolerance factor are shown in Figure 2, and it can be observed that the tolerance factor of A_2_Sn_1−x_Ti_x_Y_6_ increases monotonously with the Ti content when the A-site cation and Y-site anion are fixed. When the Ti content and the A-site are fixed, the tolerance factor increases with the decrease in the atomic radius of the Y-site anion. When the Ti content and the Y-site are fixed, the tolerance factor increases with the increase in the atomic radius of the A-site cation. However, the tolerance factor is in the range of 0.96–1.08 for all A_2_Sn_1−x_Ti_x_Y_6_ perovskites, which ensures their structural stability. 

In addition, in order to further study the thermodynamic stability of A_2_Sn_1−x_Ti_x_Y_6_ perovskite materials, the formation energies *E_f_* are obtained by the following formula [36]:(3)$$E_{f}=\left[E_{A_{2}{Sn}_{1-x}{Ti}_{x}Y_{6}}-2E_{A}-\left(1-x\right)E_{Sn}-xE_{Ti}-6E_{Y}\right]/36$$
where $E_{A_{2}{Sn}_{1-x}{Ti}_{x}Y_{6}}$ is the total energy of the primitive cell; and $E_{A}$, $E_{Sn}$, $E_{Ti}$ and $E_{Y}$ are the energies of the A, Sn, Ti and Y atoms, respectively. According to Table 1, the formation energy decreases gradually in the order of K > Rb > Cs for the same B-site cation and Y-site anion. For the same A-site and B-site cations, the formation energy increases gradually in the order of Cl < Br < I. When the A-site cation and Y-site anion are fixed, the formation energy decreases monotonously with the increase in the concentration of Ti^4+^. All A_2_Sn_1−x_Ti_x_Y_6_ perovskites exhibit negative values of formation energies, confirming that they are thermodynamically favorable, where Cs_2_TiCl_6_ has the lowest formation energy of −3.70 eV and K_2_SnI_6_ has the maximum formation energy of −2.19 eV.

### 3.2. Screening of A_2_Sn_1−x_Ti_x_Y_6_ (A = K, Rb, Cs; Y = Cl, Br, I) for Photovoltaic Applications

The band gap is an intrinsic property of materials that can not only influence the performance of the materials directly, but also provides information about the possible maximum theoretical efficiency of the materials. Thus, a critical factor for evaluating the performance of solar cells is the electronic band gap of the light absorber. The ideal band gap should be in the range of 0.9–1.6 eV [38] to obtain a theoretical efficiency higher than 25%. Especially when the band gap is around 1.3 eV, a maximum PCE in a single-junction solar cell can be achieved according to the Shockley–Queisser limit [39]. The electronic band structures of A_2_Sn_1−x_Ti_x_Y_6_ (A = K, Rb, Cs; Y = Cl, Br, I) were calculated, and the compounds all displayed semiconductor characteristic. The band structures are shown in Appendix A; Appendix A shows the band structures of K_2_Sn_1−x_Ti_x_Y_6_, Appendix A shows the band structures of Rb_2_Sn_1−x_Ti_x_Y_6_ and Appendix A shows the band structures of Cs_2_Sn_1−x_Ti_x_Y_6_. Figure 3a–c show the variation in band gap with the content of Ti in A_2_Sn_1−x_Ti_x_Y_6_ (A = K, Rb, Cs; Y = Cl, Br, I). The band gap decreases gradually in the order of Cl > Br > I for the same A-site and B-site cations, which is in agreement with the trend of band gap in MAPbX_3_ (X = Cl, Br, I) halide perovskites [40]. With the increase in the A-site cation radius, the band gap also increases for the same B-site cation and Y-site anion, indicating that the tilting of the BY_6_ (B = Sn, Ti) octahedron dictated by the A-site cation can influence the electronic structures of the material. Note that, when the Y-site is a Cl atom, the band gaps all decrease monotonically with the increase in Ti content. When the Y-site is a Br atom, the band gaps increase firstly and reach their maximum value at x = 0.5 before decreasing a little. When the Y-site is an I atom, the band gaps increase monotonically with the increase in Ti content. Based on the above calculation results, it can be observed that Ti doping plays a role in adjusting the band gap of A_2_Sn_1−x_Ti_x_Y_6_ (A = K, Rb, Cs; Y = Cl, Br, I). In addition, we found that the band gaps of A_2_Sn_1−x_Ti_x_Br_6_ (A = K, Rb, Cs) are all in the range of 1.2–1.6 eV and are very close to that of MAPbI_3_ [41,42,43], according to which the most suitable candidate for the Y-site is Br; thus, A_2_Sn_1−x_Ti_x_Br_6_ (A = K, Rb, Cs) can be considered as a promising material for photovoltaic applications.

Since the most significant role of lead-free halide double perovskite in solar cells and various opto-electronic applications is to act as a light-absorbing material, our primary focus was to find out its application in the visible and ultraviolet light absorption spectrum. The optical absorption is considered as an important property that can determine whether the material is suitable for photovoltaic applications. The absorption coefficient shows the light energy absorbed by the material, demonstrating direct transitions from the valence band maximum (VBM) to the conduction band minimum (CBM) owning to the absorption of the appropriate incident photon energies. For the sake of obtaining the optimal Ti doping concentration, Figure 3d–f show the optical adsorption spectra of A_2_Sn_1−x_Ti_x_Br_6_ (A = K, Rb, Cs) in a photon energy range from 0 to 4.5 eV. According to Figure 3d–f, the main optical absorption occurs in two energy ranges of 1.5–3.1 eV and 3.1–4.5 eV, which are mainly in the visible region and ultraviolet region, respectively. In both ranges, the replacement of the A-site cation shows a slight influence on the optical absorption, while the optical absorption intensity of A_2_Sn_1−x_Ti_x_Br_6_ (A = K, Rb, Cs) seems to be obviously influenced by the presence of the Ti dopant. With the increase in Ti content, the optical absorption coefficient increases monotonously, and when the Ti content *x* = 1, the optical absorption coefficient reaches a maximum value.

Moreover, in the range of 1.5–3.1 eV, the maximum optical absorption coefficient reaches 2.7 × 10^5^, 2.7 × 10^5^ and 2.9 × 10^5^ cm^−1^ for A = K, Rb and Cs, respectively, indicating a strong optical absorption ability in the visible light region. In the range of 3.1–4.5 eV, the Ti dopant results in a significant absorption peak in the ultraviolet region. When the Ti content is x = 0.25, the peak value is approximately the same as that in the visible region. However, the peak value is even larger than that in the visible region with the further increase in Ti content x, and the peak position shifts to the higher energy direction. The maximum optical absorption coefficients of A_2_Sn_1−x_Ti_x_Br_6_ (A = K, Rb, Cs) are as high as 4.9 × 10^5^, 4.8 × 10^5^ and 4.7 × 10^5^ cm^−1^ for A = K, Rb and Cs, respectively, indicating their significant potential to be used in ultraviolet-light-based devices. These favorable absorption properties render A_2_TiBr_6_ (A = K, Rb, Cs) the most promising candidates for photovoltaic applications. Therefore, we only focused on the properties of A_2_TiBr_6_ (A = K, Rb, Cs) hereafter in this work.

### 3.3. Mechanical, Electronic and Optical Properties of A_2_TiBr_6_ (A = K, Rb, Cs)

In order to assess the mechanical stability of the materials, it is necessary to determine the elastic constants under applied stress. The elastic constants C*ij* of A_2_TiBr_6_ (A = K, Rb, Cs) were calculated by the strain–stress relationship with finite differences implemented in the VASP code. Three independent elastic constants *C*_11_, *C*_12_ and *C*_44_ were obtained for each compound of cubic A_2_TiBr_6_ (A = K, Rb, Cs). According to Born stability conditions, all the structures of A_2_TiBr_6_ (A = K, Rb, Cs) are mechanically stable. Other mechanical constants, such as the bulk modulus, shear modulus, Young’s modulus and Poisson’s ratio, were also calculated to investigate the mechanical behavior of A_2_TiBr_6_ (A = K, Rb, Cs) by the Voigt–Reuss–Hill (VRH) approximation for polycrystals with *C_ij_*. Figure 4 shows the three-dimensional Young’s modulus and Possion’s ratio of A_2_TiBr_6_ (A = K, Rb, Cs). Both the Young’s modulus and Possion’s ratio decrease with the increase in the A-site ionic radius.

Figure 5 shows the mechanical properties of A_2_TiBr_6_ (A = K, Rb, Cs). It can be found that the elastic constants *C*_11_, *C*_12_ and *C*_44_ decrease monotonously with the increase in the ionic radius of the A-site. K_2_TiBr_6_ has the highest *C_ij_* and Cs_2_TiBr_6_ has the lowest *C_ij_*. The increase in the ionic radius of the A site plays a more important role in decreasing the bulk modulus B and Young’s modulus E than in decreasing the shear modulus G. The Pugh’s ratio B/G and Poisson’s ratio are two indicators that are usually used to evaluate the ductile or brittle behavior of solid materials. If the Pugh’s ratio is lower than 1.75, the material is considered to be brittle; otherwise, it is ductile. For the Possion’s ratio, if its value is higher than 0.26, the material is considered to be ductile; otherwise, it is brittle. From Figure 5, it can be seen that the A_2_TiBr_6_ (A = K, Rb, Cs) perovskites are all ductile materials; thus, they are candidates of flexible photovoltaic devices. Since lower Pugh’s ratio and Possion’s ratio values mean a higher hardness and vice versa, the hardness of A_2_TiBr_6_ (A = K, Rb, Cs) is in the order of K_2_TiBr_6_ < Rb_2_TiBr_6_ < Cs_2_TiBr_6_.

Figure 6 shows the band structures and the density of states of A_2_TiBr_6_ (A = K, Rb, Cs). It can be observed that the band gap slightly increases with the increase in the A-site cation radius, which is 1.38, 1.43 and 1.52 eV, for K_2_TiBr_6_, Rb_2_TiBr_6_ and Cs_2_TiBr_6_, respectively. In order to better understand the origin of the conduction band and valence band, the density of states (DOS) and partial DOS of A_2_TiBr_6_ (A = K, Rb, Cs) were plotted. It can be seen that the Br-*4p* orbitals mainly contribute to the VBM, while the CBM receives its primary contribution from the Ti-3d orbital and a very small amount from the Br-*4p* orbital. The A-site cations in A_2_TiBr_6_ (A = K, Rb, Cs) contribute little to the band edges, indicating no direct influence on the band gaps.

In order to further check the orbital contribution, the decomposed charge density distribution of VBM and CBM were calculated and are plotted in Figure 7. It can be seen that the charge density of VBM is distributed on all Br atoms composed of antibonding state of Br-*4p* and Br-*4p*, while the charge density of CBM is mainly distributed on Ti atoms. Both the partial DOS and the decomposed charge density confirm the negligible contribution of the A-site to the band edge and, as a result, there is no direct influence on the band gap.

To confirm the possible application of A_2_TiBr_6_ (A = K, Rb, Cs) as a photovoltaic material, the various optical parameters of A_2_TiBr_6_ (A = K, Rb, Cs) were calculated for the incident energy range of 0–25 eV, as shown in Figure 8. The dielectric constant is the most important parameter to illustrate the optical behavior of the material and it can be used to calculate several parameters. The real part of the dielectric constant ε_1_(ω) shows the polarization and dispersion of impinging light, and the static dielectric constant ε_1_(0) is 3.67, 3.44 and 3.17 for K_2_TiBr_6_, Rb_2_TiBr_6_ and Cs_2_TiBr_6_, respectively, according to Figure 8a, which are inversely related to the respective band gaps. The imaginary part ε_2_(ω) demonstrates the light absorption with the maximum absorption peaks at around 1.9–2.0 eV for A_2_TiBr_6_ (A = K, Rb, Cs), as shown in Figure 8b. The absorption coefficient measures the ability of the materials to absorb light energy per centimeter. Figure 8c shows the absorption spectrum of A_2_TiBr_6_ (A = K, Rb, Cs), in which the absorption edges are very important to indicate the critical points after which the absorption linearly increases, and for A_2_TiBr_6_ (A = K, Rb, Cs), the absorption edges are all in the infrared energy range. In the visible energy range (1.77–3.18 eV), the dominant absorption peak shifts to a higher energy direction when A is from K to Cs. The refractive index is the ratio of the speed of light in vacuum to the speed of light in the material, often used to measure the transparency of materials and it provides details similar to the real part of the dielectric constant; thus, both curves display similar trends. The calculated static refractive index *n*(0) for A_2_TiBr_6_ (A = K, Rb, Cs) was 1.91, 1.85 and 1.78, respectively, as shown in Figure 8d. Since the calculated static dielectric constant is 3.67, 3.44 and 3.17, it is evident that these two parameters follow the following relation:(4)$$n(\omega )=\sqrt{\varepsilon_{1}(\omega )}$$

The reflectivity shows the ability of the material to reflect the incident energies to its surface, and the static reflectivity R(0) of A_2_TiBr_6_ (A = K, Rb, Cs) is 9.8, 9.0 and 7.9%, respectively, according to Figure 8e, which can be used to evaluate the surface quality. The maximum reflectivity is obtained as 31.9, 25.1 and 30.8% at 8.9, 9.2 and 7.4 eV, respectively. The loss factor L(*ω*) shows the energy loss due to the dispersion and scattering of light, usually representing the interband, intraband and plasmonic interactions, which is displayed in Figure 8f. The energy loss occurs when electrons move at a high speed through the material and scatter, and reaches its maximum value when the energy of the incident light is approximately 9 eV. Since the band gaps of A_2_TiBr_6_ (A = K, Rb, Cs) are well below the energies where electron energy losses are maximum, A_2_TiBr_6_ (A = K, Rb, Cs) are potential materials for photovoltaic applications.

## 4. Conclusions

In summary, we predicted a series of vacancy-ordered, lead-free, Ti-based all-inorganic double perovskites for photovoltaic applications by density functional theory. The comprehensive theoretical investigation of the structural stability, mechanical, electronic and optical properties of A_2_Sn_1−x_Ti_x_Y_6_ (A = K, Rb, Cs; Y = Cl, Br, I) was performed. The calculated tolerance factors of A_2_Sn_1−x_Ti_x_Y_6_ (A = K, Rb, Cs; Y = Cl, Br, I) are in the range of 0.96–1.08 for all compositions, which ensures their structural stability. The calculated negative formation energies confirm they are also themodynamically favorable. The band structure and optical absorption results show that A_2_Sn_1−x_Ti_x_Br_6_ (A = K, Rb, Cs) compounds all show semiconductor characteristics with band gaps in the range of 0.9–1.6 eV, and A_2_TiBr_6_ (A = K, Rb, Cs) compounds have the maximum optical absorption both in the visible and ultraviolet energy ranges; thus, they can be considered as promising materials for photovoltaic applications. For A_2_TiBr_6_ (A = K, Rb, Cs) compounds, they are all ductile perovskites with mechanically stable structures. The VBM of A_2_TiBr_6_ (A = K, Rb, Cs) receives its mainly contribution from the Br-*4p* orbitals and the charge density of VBM is distributed on all Br atoms composed of the antibonding state of Br-*4p* and Br-*4p*. The CBM of A_2_TiBr_6_ (A = K, Rb, Cs) receives its primary contribution from the Ti-3*d* orbital and a very small amount from the Br-*4p* orbital, and the charge density of CBM is mainly distributed on Ti atoms. This work can provide theoretical guidance for further theoretical research as well as experimental research on the application of A_2_TiBr_6_ (A = K, Rb, Cs) in photovoltaic and other optoelectronic applications.

## Figures and Tables

**Figure 1 nanomaterials-13-02744-f001:**
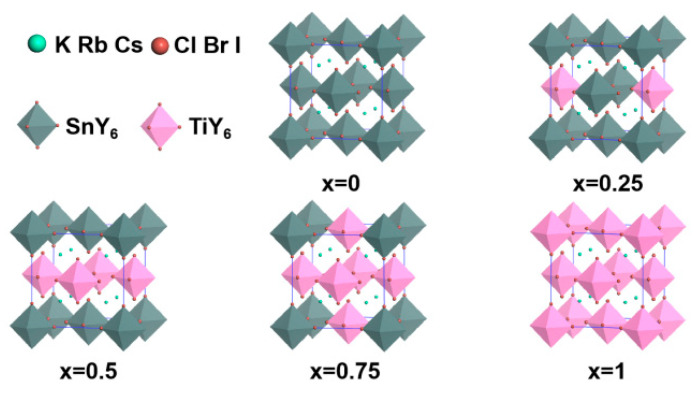
Optimized crystal structures of A_2_Sn_1−x_Ti_x_Y_6_ (A = K, Rb, Cs; Y = Cl, Br, I).

**Figure 2 nanomaterials-13-02744-f002:**
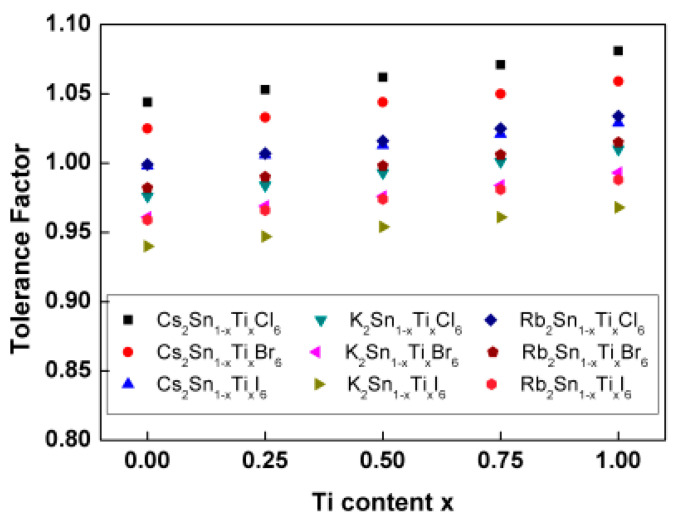
Tolerance factor of A_2_Sn_1−x_Ti_x_Y_6_ (A = K, Rb, Cs; Y = Cl, Br, I).

**Figure 3 nanomaterials-13-02744-f003:**
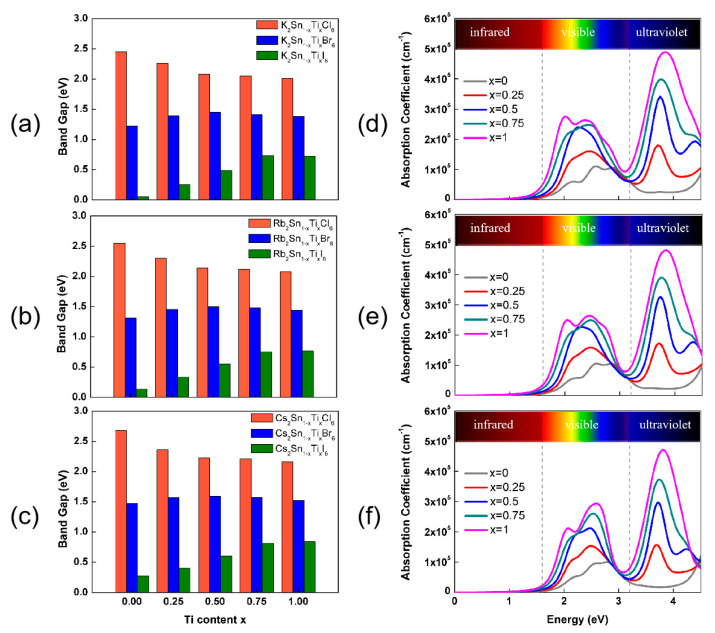
Band gaps of A_2_Sn_1−x_Ti_x_Y_6_ (A = K, Rb, Cs; Y = Cl, Br, I) versus Ti content x: (**a**) K_2_Sn_1−x_Ti_x_Y_6_, (**b**) Rb_2_Sn_1−x_Ti_x_Y_6_ and (**c**) Cs_2_Sn_1−x_Ti_x_Y_6_. Optical absorption spectra of A_2_Sn_1−x_Ti_x_Br_6_: A = K in (**d**), A = Rb in (**e**) and A = Cs in (**f**).

**Figure 4 nanomaterials-13-02744-f004:**
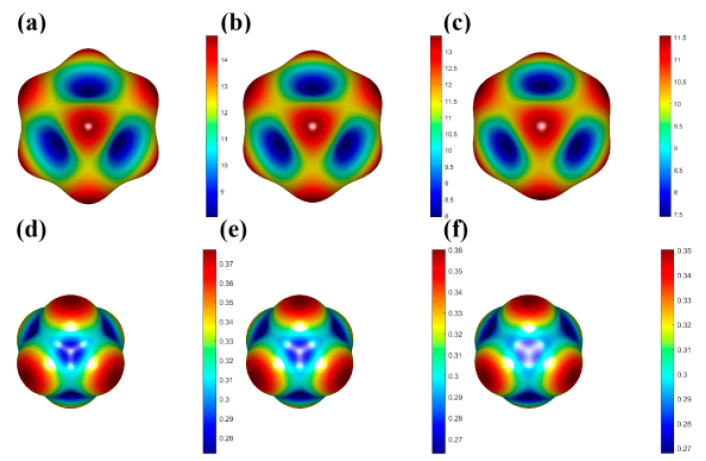
Three-dimensional Young’s modulus of (**a**) K_2_TiBr_6_, (**b**) Rb_2_TiBr_6_ and (**c**) Cs_2_TiBr_6_ and Possion’s ratio of (**d**) K_2_TiBr_6_, (**e**) Rb_2_TiBr_6_ and (**f**) Cs_2_TiBr_6_.

**Figure 5 nanomaterials-13-02744-f005:**
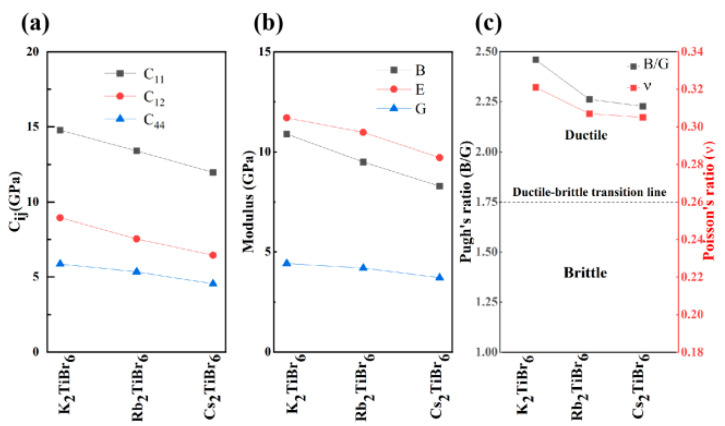
Mechanical properties of (**a**) K_2_TiBr_6_, (**b**) Rb_2_TiBr_6_ and (**c**) Cs_2_TiBr_6_.

**Figure 6 nanomaterials-13-02744-f006:**
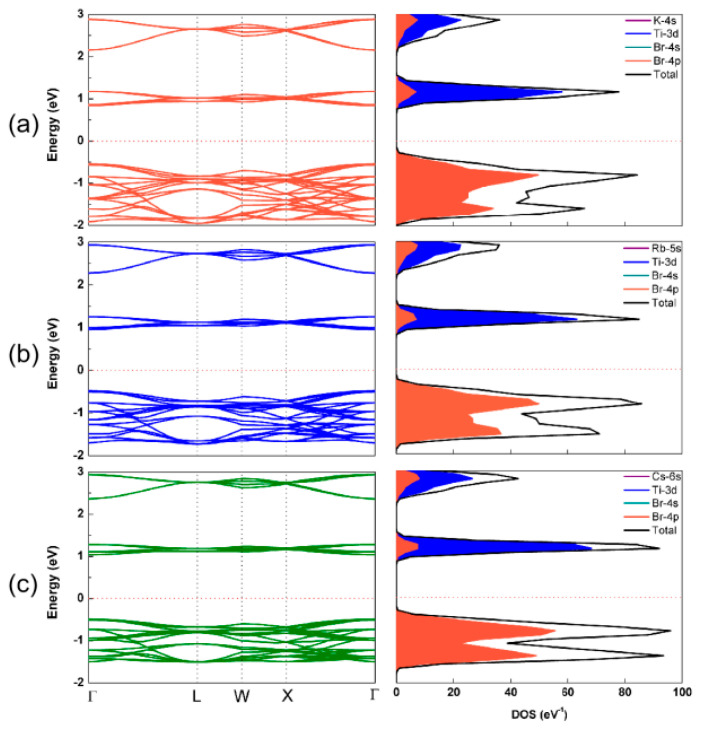
Band structures and density of states of (**a**) K_2_TiBr_6_, (**b**) Rb_2_TiBr_6_ and (**c**) Cs_2_TiBr_6_.

**Figure 7 nanomaterials-13-02744-f007:**
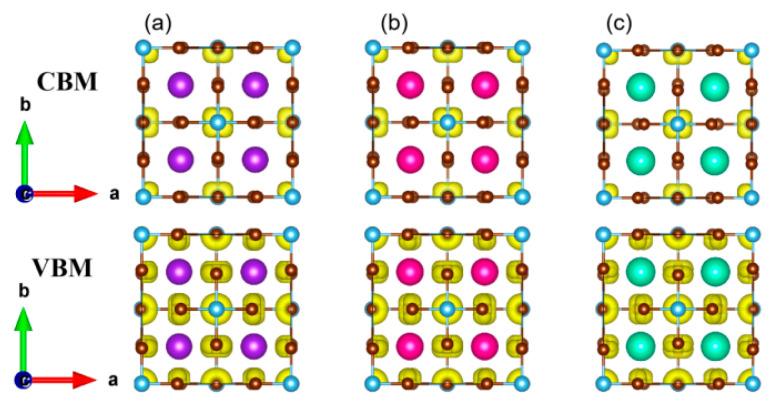
Charge density of CBM and VBM of (**a**) K_2_TiBr_6_, (**b**) Rb_2_TiBr_6_ and (**c**) Cs_2_TiBr_6_.

**Figure 8 nanomaterials-13-02744-f008:**
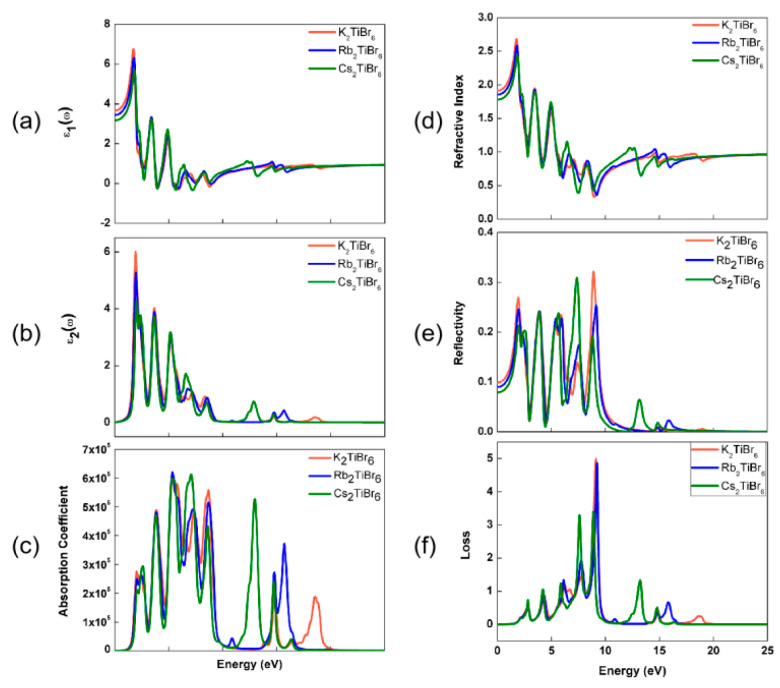
Calculated optical parameters of A_2_TiBr_6_ (A = K, Rb, Cs): (**a**) real dielectric constant, (**b**) imaginary dielectric constant, (**c**) absorption coefficient, (**d**) refractive index coefficient, (**e**) reflectivity coefficient and (**f**) loss coefficient.

**Table 1 nanomaterials-13-02744-t001:** The formation energy of A_2_Sn_1−x_Ti_x_Y_6_ (A = K, Rb, Cs; Y = Cl, Br, I).

*x*	0	0.25	0.5	0.75	1
KSn_1−x_Ti_x_Cl_6_	−2.99	−3.16	−3.34	−3.51	−3.68
KSn_1−x_Ti_x_Br_6_	−2.61	−2.77	−2.93	−3.09	−3.26
KSn_1−x_Ti_x_I_6_	−2.19	−2.34	−2.49	−2.64	−2.79
RbSn_1−x_Ti_x_Cl_6_	−3.00	−3.17	−3.34	−3.51	−3.69
RbSn_1−x_Ti_x_Br_6_	−2.62	−2.78	−2.94	−3.10	−3.26
RbSn_1−x_Ti_x_I_6_	−2.20	−2.35	−2.50	−2.65	−2.80
CsSn_1−x_Ti_x_Cl_6_	−3.01	−3.19	−3.36	−3.53	−3.70
CsSn_1−x_Ti_x_Br_6_	−2.64	−2.80	−2.96	−3.12	−3.29
CsSn_1−x_Ti_x_I_6_	−2.24	−2.39	−2.53	−2.68	−2.83

## Data Availability

Not applicable.

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
