# Peer review of "Investigation of Vacancy-Ordered Double Perovskite Halides A2Sn1−xTixY6 (A = K, Rb, Cs; Y = Cl, Br, I): Promising Materials for Photovoltaic Applications"

_nanomaterials, 2023, doi:10.3390/nano13202744_

Round 1

Reviewer 2 Report

I read the manuscript "Investigation of vacancy-ordered double perovskite halides A2Sn1-xTixY6 (A=K, Rb, Cs; Y=Cl, Br, I): promising materials for photovoltaic applications" and did not find significant problems. I hope the authors will make this manuscript more friendly to experimentalists before recommending publication. Please see the following points.

1. The authors should compare their calculated results with the reported experiments. The optical properties will be easy for such comparison.

2. How about the case of A = Li, Na or Y = F, (At)? 

3. Appendices A and B should be removed because there is no information here.

Round 2

Reviewer 1 Report

I am satisfied with the Authors' responses to my queries. I suggest the acceptance of the revised manuscript in its current form.